# How Does Regional Innovation Capacity Affect the Green Growth Performance? Empirical Evidence from China

**Sumin Hu [1], Shulin Liu [1], Die Li [1],\* and Yuxuan Lin [2]**

[1]   School of Economics, Wuhan University of Technology, Wuhan 430070, China
[2]   W.P. Carey School of Business, Arizona State University, Tempe, AZ 85281, USA
\*   Correspondence: lidie1004@whut.edu.cn; Tel.: +86-1582-748-9506

**Abstract:** Behind the high development of technology, backward institution systems and imperfect incentive mechanisms are not conducive to the green transformation of the economic society in China. Meanwhile, the relative effectiveness of both technical and institution innovation in encouraging green growth has yet to be tested empirically in China. It is of great practical significance to assess the effect of regional innovation capacity (RIC) on the green growth performance. This paper firstly exploits a model to measure regional innovation capacity from the perspective of technological and institutional respect. The panel data of 30 provinces in China during 2008–2017 is then used to examine the coordination effect of technological and institutional instruments on green growth performance. The empirical results demonstrate the following: (i) regional innovation capacity significantly affects the green growth performance of 30 provinces in China, showing regional differences. The elasticity of RIC on the green total factor efficiency in the eastern region is larger at approximately 0.48, followed by central and western areas, at about 0.47 and 0.45, respectively; (ii) technological innovation is able to incentivize green growth performance for all regions in China, while the institutional innovation induces green growth in the eastern region only; (iii) the coordination of technical and institutional instruments has a significant effect on green growth performance, positive in the eastern region and negative in central region respectively.

**Keywords:** regional innovation capacity; green growth performance; technological innovation; institutional innovation; coordination level

## 1. Introduction

Green growth has attracted widespread attention over the last two decades, especially in China. It is believed that the extensive industrial growth model of China has already led to substantial consumption of resources and ecological deterioration of the environment. To cope with this situation, given that a successful transformation towards a green economy involves complex interactions between economic actors [1,2], the development of green and sustainable economy has recently become an important subject of policymakers' concern. Thus, the innovation system (IS) concept is used in this green and sustainable transition, which emphasizes the importance of positive interactions among a variety of actors, including firms, universities, policy makers, and various intermediaries under specific institutional contexts [3,4].

Since the pioneering work of Freeman (1987) in analyzing the essential role of national innovation systems in economic growth, the innovation system has become an important framework for studying national innovation capability and green growth in recent years [5,6]. For a large developing country like China, it may be inappropriate to analyze its innovation system at the national level because of the

diversity of its internal differences among regions and industries [7]. As such, it is of great value to analyze the impact of innovation system on green growth at the regional level. However, the specific definition and connotation of "system" and its boundaries are still vague and unclear, which brings many difficulties for empirical analysis within the framework of innovation system.

Regional innovation system studies have always focused on specific innovation capacity and its impact on sustainable development and industrial transition [8,9]. The mainstream of previous literature in the green growth domain focuses its attention on the technological aspect of regional innovation capacity (RIC) [10–12], while recent studies pay more attention to the integration of various social aspects or criteria in the green growth process [13], specifically the role of innovation system in the development of green growth, which can be captured by the notion of RIC. Although regional innovation capability is endowed with different connotations in different institutional backgrounds and periods, both external forces and internal resources are required in the building of regional innovation capacity [14,15]. This calls for the coordination of both technological approach and institutional approach for actors to deal with regional resources and environmental constraints.

However, according to a recent published report, China's innovation index ranks 17th in the world in 2018, and 70th in the sub-item of institutional environment (The report of Global Innovation Index 2018 published jointly by WIPO, Cornell University and the European School of Business Administration). This suggests that China's institutional construction lags behind technological development of green technology domain. Behind the high development of technology, backwards institution systems and imperfect incentive mechanisms are not conducive to the green transformation of the economic society in China. In addition, the relative effectiveness of both technical and institutional innovation in the process of green growth transition has not been empirically tested. It is of great practical significance to assess the effect of regional innovation capacity on green growth in China.

While technological innovation or institutional innovation capacity as the decisive factor of economic growth has been debated for a long time, previous literature has mainly focused its attention on the essential role of institutional factors in promoting economic development [14,16,17]. Especially in the context of China's economic system transition, institutional innovation is more urgent than technological innovation [18,19]. These studies highlight the importance of institutional factors in economic development. However, recent literature pays more attention to the common role of technology and institution in promoting economic development [20–22]. The present studies emphasize the key role of institutional innovation in green economic growth from theoretical and empirical levels. How the relationship between theories and empirical practices affects green economic development is not explored to the authors' understanding. To address this problem, this paper exploits a model to measure regional innovation capacity from the perspective of technological and institutional respect. The coordination effect of technological and institutional instruments on the green growth performance is examined.

Empirical studies which focus on the impact of regional innovation capacity on green growth has attracted widespread attention from academia in recent years [23–25]. We extend the literature by exploiting an econometric assessment of China's regional innovation capacity in respect of both technological and institutional concern. The effect of green economy growth on the regional innovation capacity of China's provinces is tested by using China's provincial data on regional innovation capacity. The target is primarily the industrial enterprises above scale: colleges, universities, and various R&D institutions. Results show that regional innovation capacity led to a significant increase in the green economy growth between 11% and 93%.

The next section provides a literature review as well as the principal hypotheses on the role of regional innovation capacity in promoting green and sustainable development. The third section defines the data set, the operationalization of regional innovation capacity variables including technological innovation capacity, institutional innovation capacity, and the coordination feature of these two innovation capacities in China. Whereas the fourth section presents the the model specification

and empirical results, finally, the last section concludes with a summary of the main results, which highlights the policy implications and outlines possible further research directions.

## 2. Literature Review and Research Hypotheses

The main objective of this paper is to test the effect of two types of regional innovation capacity (RIC) on green and sustainable economic development. Previous studies have explored various factors affecting green growth from different perspectives [26–29]. With the rise of national and regional innovation system theory in the 1990s, there is a growing attention of literature on the role of regional innovation capacity. The regional innovation capacity has been found to be of equal importance as the green growth drivers [30].

In the early work of Freeman [6] and Nelson [2], institutional structure and cultural differences in innovation are the key factors contributing to the national differences in innovation performance. Since then, RIC scholars have developed a series of theories and empirical strategies related to regional differences and socio-technical innovation from the perspective of earlier static analysis to current dynamic approach [4]. These studies highlight both technological advance and institutional structure as vital roles in the green transition processes, although neoclassical economics [31] and new institutional economics [32] have contributed to finding out which forces drive green growth and consequently adopt different emphases on the roles of technology and institution in green growth, respectively [14,33].

On the technical side, previous studies suggest that research and development (R&D) activities are very important innovation instruments in shaping regional technical innovation capacity [34,35]. It is well-known that R&D activities are highly uncertain, indivisible, and non-exclusive, which may lead to market failure and insufficient R&D investment, especially in basic research fields. Thus, the production and diffusion of knowledge and innovation is efficient only when government, research institutions, and enterprises interact in a constructive, interactive, and complementary way [36]. In this respect, public policies are likely to provide a favorable situation for technical innovation and capacities acquired by R&D activities. Therefore, R&D policy instruments have been widely used in the broad analytic framework of RIC over the years [37,38]. With respect to green growth, public subsidy mechanism plays a vital role in inducing ecological innovation and improving regional innovation capacities, which consequently reduce the cost of pollution and emission abatement. The existing works of literature discusse the role of technological innovation in green growth mainly on specific technological fields, such as technological upgrading of new energy automotive industry, improvement of clean energy productivity [39–41]. These studies emphasize that technological innovation capacity is an effective means to promote regional green economy development in the long run [24,42,43].

With focus on the development of a regional green economy, these analyses are expanded by studying the impact of regional technological innovation capability and its coordination characteristics of green growth performance. Since technical progress is the basis for the green economic growth, it is expected that regional technological innovation capacity fosters inventive activity and thereby increases the performance of green growth.

The first hypothesis proposed in this paper is stated as follows:

**Hypothesis 1 (H1).** *Regional technological innovation capacity increases the performance of green growth.*

With respect to the institution sides, a growing number of recent studies pay attention to the role played by institutional factors in fostering green/clean technology in the context of market failures and environmental externality [4,24,44]. For example, the importance of the technology policy is highlighted in [28,45], and the significance of the environmental policy is discussed in [46,47]. These studies highlight the primary role played by RIC in driving green growth. They also suggest that the perfection of the institutional mechanism is of great significance to innovation [1,4]. Acemoglu et al. examines how institutional change affects the development of a country or region. It is found that the fundamental institutional changes brought about by the French Revolution led to the long-term economic growth in

the occupied areas [48]. Kirchherr et al. points out that institutional structure and cultural difference are likely to be circular economy barriers rather than a single technological barrier [49].

The constructions of regional, institutional, and systemic innovation capacity aim to connect various inventive actors, such as enterprises, research colleges, and relevant institutes. Their inventive cooperation is fostered through learning, technology, and resource sharing process among those parties [19,28]. Those constructions include public education and information support instruments, market system support instruments, financial support instruments, and government policy support instruments. They help to shape green growth by providing infrastructure, building knowledge exchange, and technology transfer platforms. Facilitating knowledge learning and exchange and enhancing cooperation is expected [29]. There is extensive empirical literature on the role of institutional instruments in shaping green growth. Wu et al. shows that the roles and positions of the universities are the most important in the cooperative of the industry–university–research institution, which is one of the main ways for enterprises to gain competitiveness [50]. Using China's provincial data, Liu et al. show that the perfection of the financial system has a significant effect on green economic growth. They pointed out that the positive role of the stock market is stronger than that of the banking sector. In other words, the market-oriented financial structure system is more conducive to the long-term growth of China's green economy [30]. These institutional instruments provide various public support for promoting the green economy. Furthermore, by providing various incentives and improving the institutional environment, it is possible to increase the performance of green growth by reducing the risk of innovation investment and forming cooperation with more potential partners. Thus, the second hypothesis in this paper is highlighted as follows:

**Hypothesis 2 (H2).** *Regional institutional innovation capacity increases the performance of green growth.*

Although the instruments mentioned above seem to be related to the increase in green innovation activities, most of them focus on the distinct role of innovation instruments in shaping regional innovation capacity. While they are frequently implemented simultaneously in practice, it is necessary to constitute such a mix of instruments to increase inventive activity, especially for green/clean innovation. Michael P. Todaro believes that economic development should be viewed as a multidimensional process including the restructuring and restructuring of the entire economic and social system [51]. However, a variety of empirical studies isolate technological approach and institutional approach in their analytic framework, and they do not explore how the interaction between them affects green growth. The intrinsic motive force of economic development depends on technological progress and the construction of a corresponding institution structure, which is the process of coordination and unification of technology and institution; Furman further pointed out that the key to building an innovation-driven system is to maintain and coordinate the relationship between these factors [52].

Since the Reform and Development of China, there it has experienced continuous optimization of the institutional environment and the upgrading of technological level. Both technology and institutions are likely to contribute to China's green growth performance and total factor productivity (TFP). One of the key factors in the sustainability of technological innovation is whether the current institutional arrangements provide suitable conditions for the occurrence and diffusion of technological innovation. Another is whether there is an effective property rights system that stimulates innovation and reduces risks to innovation, including private property rights, patent systems, and intellectual property protection systems. These are key factors to the sustainable development of technological innovation. Based on the existing research findings, it is suggested that both technological and institutional instruments should be included in measuring regional innovation capacity. In addition, the role of the coordination level between technological and institutional innovation in promoting green growth was tested as well. It is expected that both technology and institution create incentives. Thus, the following hypothesis in this paper is highlighted:

**Hypothesis 3a (H3a).** *The performance of green growth is significant and positively affected by the combination of technological instruments and institutional instruments.*

**Hypothesis 3b (H3b).** *The level of coordination between regional technological innovation capacity and institutional innovation capacity has a positive effect on green growth performance.*

According to the above discussion and the conceptual model presented in Figure 1, the three kinds of RIC directly affect economic growth. The effects from the developing green and clean technology are realizing the recycling of resources, optimizing the energy consumption structure, and improving the utilization rate of resources. While institution and systemic innovation aims to connect different innovators, such as universities, scientific research institutions, and enterprises and encourage knowledge and technology learning and resource sharing by building knowledge exchange and technology transfer platforms, it includes infrastructure construction and encourages various forms of technical cooperation among innovators [53,54]. The combination of these two types of innovation capacity constitutes the regional innovation system, which needs to be consistent to maintain the long-term sustainability of green growth.

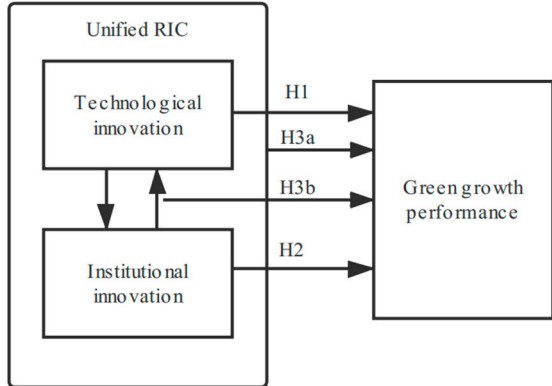

**Figure 1.** Hypothesis of RIC on green growth performance.

## 3. Regional Innovation Capacity and Coordination Characteristics

A specific effort to map regional innovation capacity, as well as the coordination of technological innovation and institutional innovation in the field of green growth, is proposed in this section. In order to empirically test the regional innovation capacity as well as the coordination of technological innovation and institutional innovation in the field of green growth, the information on the two alternative types of innovation capacity related to the promotion of green growth performance in each province of China is retrieved. Then, we focus on the coordination of technological and institutional innovation in the regional system. The impact mechanism on the green growth is further analyzed by constructing an economic measurement model of coordination between technological innovation and institutional innovation.

### 3.1. Regional Innovation Capacity

The regional innovation capacity in the technological and institutional respects is explored. Firstly, we consider the regional technical innovation capacity as for the technology-support for green growth. Secondly, we consider the institution system as for the institution-support for green growth. Our indicator representing technical innovation capacity is based on Knowledge Creation Function [25,55] and National Innovation System Theory [6]; the first type includes innovation resources [39], knowledge creation capacity [56], and enterprises' innovation capacity instruments. It aims to provide technical and knowledge support for green growth [8]. Following the approach proposed by Nelson (1987) [57] and Liu (2019) [30], the second type are institutional and systemic instruments including education

and information, market institution, financial support, and government support. It aims to reinforce the institutional support and consequently achieve green growth targets [35]. Detail information of these variables are described in Table 1.

**Table 1.** Regional innovation capacity and instruments.

| Type | Support Type | Instruments | Weight |
|------|--------------|-------------|--------|
| Technical innovation capacity | Innovative resources | R&D personnel full-time equivalent per 1000 people | 0.037603 |
| | | R&D expenditure to GDP | 0.026020 |
| | | College students per 100,000 people | 0.020177 |
| | Knowledge creation | Number of SCI papers/researchers | 0.030449 |
| | | Total patent authorizations/R&D personnel | 0.070025 |
| | | Invention Patent authorizations/Total patent authorizations | 0.023595 |
| | Firm innovation | R&D personnel full-time equivalent in firms per 1000 people | 0.045251 |
| | | Enterprise R&D Expenditure to GDP | 0.023562 |
| | | Technology market transactions to GDP | 0.104054 |
| | | Trademarks per capita | 0.067093 |
| | | Total import and export of high-tech products to GDP | 0.051567 |
| | | Sales revenue of new products/industrial enterprises main business income | 0.030430 |
| Institutional innovation capacity | Education and information | Science and technology libraries per capita | 0.029398 |
| | | Education expenditure to GDP | 0.066295 |
| | | Computers owners and broadband users per 1000 people | 0.028395 |
| | Market institution | Foreign direct investment to GDP | 0.074983 |
| | | The proportion of investment in fixed assets of non-state-owned enterprises | 0.013156 |
| | | The proportion of industrial output value of non-state-owned enterprises | 0.012466 |
| | Financial support | Loan Balance of Financial Institutions to GDP | 0.029071 |
| | | Total Capital Formation in Stock Market to GDP | 0.055474 |
| | Policy support | Government R&D input to GDP | 0.087072 |
| | | Public science and technology expenditure to GDP | 0.040275 |
| | | Public education expenditure to GDP | 0.033588 |

Technique for Order Preference by Similarity to an Ideal Solution (TOPSIS) is used to measure the comprehensive level of regional policy support in China from 2008 to 2017. According to the proximity of each evaluation unit and the understanding image, it is sorted according to the size. The average value is between 0 and 1. The closer it is to 1, the higher the evaluation objective level is. The combination of the two methods can overcome the influence of subjective factors and make the results more convincing. It also helps to understand the relative advantages and disadvantages of technological innovation policies. Accordingly, measuring of innovation capacity is as follows: (1) Normalization of index matrix. By using range standardization method, the indicators in this paper are all positive indicators. $X'_{ij} = \frac{X_{ij} - \min(X_{ij})}{\max(X_{ij}) - \min(X_{ij})}$, where $X_{ij}$ and $X'_{ij}$ denote the government support level measurement index of province $i$ and policy $j$ before and after standardization respectively; (2) Determining information entropy: $f_{ij} = X'_{ij} / \sum_{i=1}^{n} X'_{ij}$ and $H_j = \ln \frac{1}{n} \sum_{i=1}^{n} (f_{ij} \ln f_{ij})$, where $f_{ij}$ denotes the characteristic weight of policy $j$ in province $i$; (3) Weight of Measuring Indicators: $W_j = (1 - H_j) / \sum_{j=1}^{m} (1 - H_j)$; (4) Determining positive and negative ideal solutions based on weighted matrix: $S_j^+ = \sum \max(\gamma_{ij})$ and $S_j^- = \sum \min(\gamma_{ij})$, where $\gamma_{ij} = W_{ij} \times X'_{ij}$; (5) Computing the closeness between the evaluation unit and the optimal ideal solution as follows:

$$C_i = \frac{\sqrt{\sum_{j=1}^{m}(S_j^- - \gamma_{ij})^2}}{\sqrt{\sum_{j=1}^{m}(S_j^+ - \gamma_{ij})^2} + \sqrt{\sum_{j=1}^{m}(S_j^- - \gamma_{ij})^2}} \tag{1}$$

where $\sqrt{\sum_{j=1}^{m} \left(S_j^+ - \gamma_{ij}\right)^2}$ and $\sqrt{\sum_{j=1}^{m} \left(S_j^- - \gamma_{ij}\right)^2}$ denotes the Euclidean distance of measured unit between positive ideal solution and negative ideal solution respectively. According to the degree of closeness, each evaluation unit is ranked. The higher the value of $C_i$, the higher the ranking, which captures RIC and its sub-items (technological innovation and institutional innovation).

### 3.2. Characterizing Regional Innovation Capacity

Although the $C_i$ value above reflects regional innovation capability and its level, it does not fully reflect how technological innovation and institutional innovation affect the process of regional green growth. A large number of empirical studies on innovation-driven approaches show that regional green growth is a dynamic process of continuous interaction between institutional innovation and technical innovation. The coordination of both technological and institutional instruments are the key factors to achieve green growth performance.

Therefore, this paper pays more attention to the coordination of technical and institutional innovation capacity in regional systems. The coordination of regional innovation capability emphasizes that the development of different innovation motives within the region should be coordinated and balanced, otherwise it will hinder the process of green growth. Based on the original contributions of Frenken et al. (2007) [58] and Los and Timmer (2005) [59] on technology proximity matrix, this paper constructs a coordination measurement model between technological innovation capability and institutional innovation capability as follows:

$$Psim_{it} = \left[ \frac{|Tech_{it} - Inst_{it}|}{\sqrt{Tech_{it} - Inst_{it}}} \right]^{-1}, i = 1, \ldots, N \tag{2}$$

where $Tech_{it}$ and $Inst_{it}$ represents regional technological innovation capacity and institutional innovation capacity respectively; the value of $Psim_{it}$ is expected to capture the coordination of technical and institutional capacity by measuring the similarity of the two innovation-related instruments. The larger the value of $Psim_{it}$, the greater the level of coordination between them. Consequently, it will have a much more positive effect on green growth.

### 3.3. Analysis on the Evolution Trend of RIC

#### 3.3.1. Regional Innovation Capacity

According to Equation (1), the unified regional innovation capacity (Unified), technology innovation capacity (Tech), Institutional innovation instruments (Soft) are determined. Figures 2–4 visually present the average unified capacity, technology capacity, and institutional capacity in different regions of China over the period 2008–2017. Which they suggest that the three innovation capacities are generally low in China (less than 0.40). As shown, the trend is almost the same in these three types of RIC, while there is sharp rise in soft curve after 2014 and a dramatic fall in 2015.

Additionally, the differences among Unified, Tech, and Soft are small in the central region and eastern region, while greater in western region. The differences are smallest in the eastern region. This region also has a much higher value. While provinces with great differences among these curves show a relative lower value of the innovation capacity. This suggests that these developed regions with higher economic level are likely to pay more attention to the development of technology as well as the improvement of soft and system institution. There is still considerable room for improvement of innovation capacity in the central part and western areas.

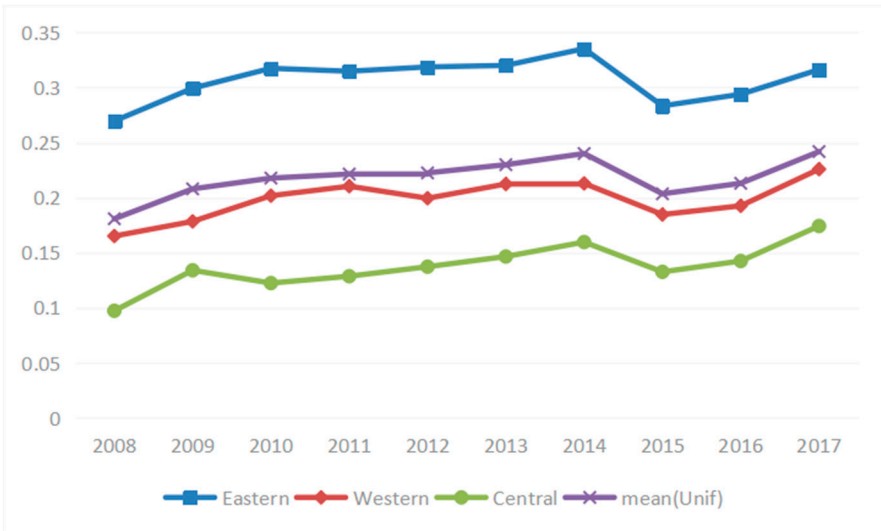

**Figure 2.** The average regional innovation capacity (Unified) from 2008–2017. Source: Author's calculations on China's Year Book.

Figure 2 directly shows the regional differences of unified innovation capacity. It shows that the differences among eastern provinces, central provinces, and western provinces are obvious in unified RIC. The unified RIC is above average level in the eastern region and is much higher in the western region and the central region.

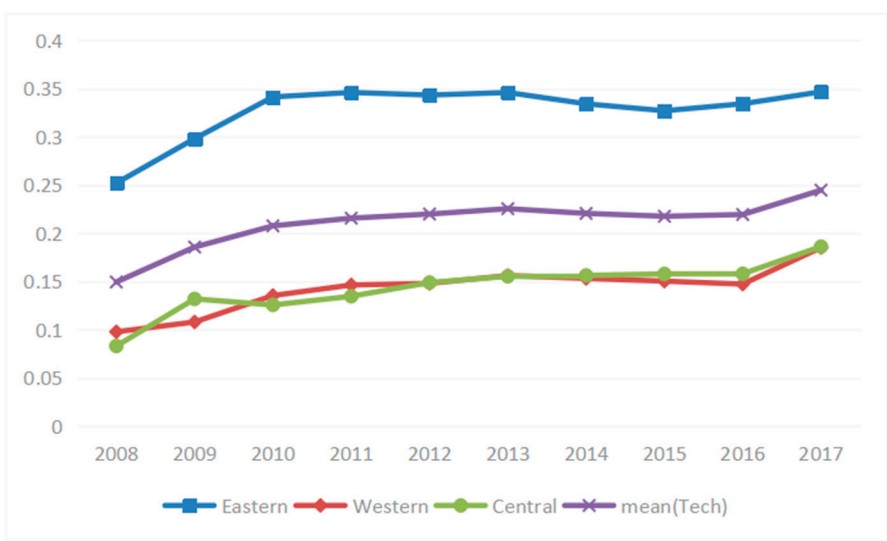

**Figure 3.** The average regional innovation capacity (Tech) from 2008–2017. Source: Author's calculations on China's Year Book.

Figure 3 shows the regional differences of technological innovation capacity. It can be found that the differences among eastern provinces, central provinces, and western provinces are obvious in technical RIC, which is above average level in eastern region, followed by western region and central region.

Figure 4 reports the regional differences of institutional innovation capacity. It shows that the differences among eastern provinces, central provinces, and western provinces are obvious in institutional RIC. The eastern region shows a much higher value, followed by an average value in the western region and central region.

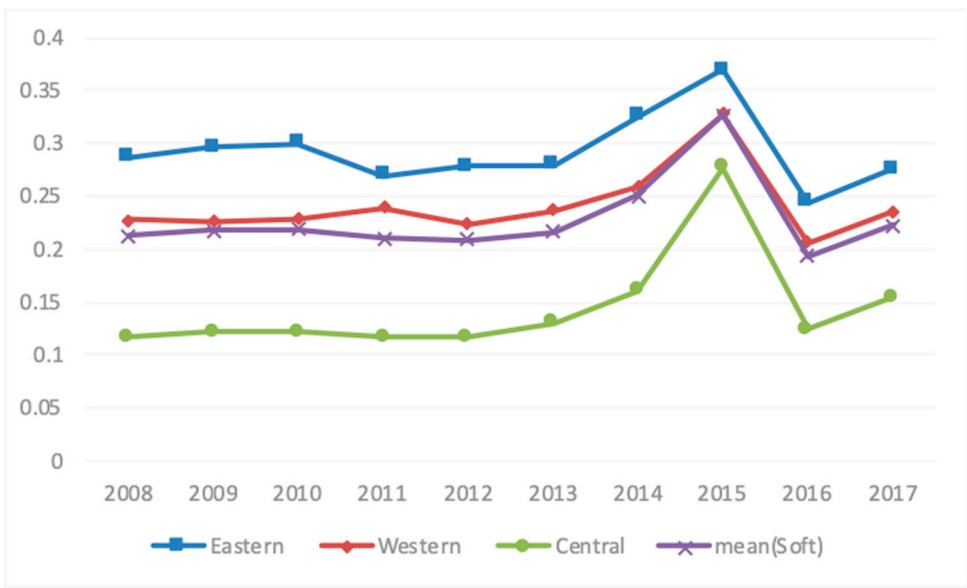

**Figure 4.** The average regional innovation capacity (Soft) from 2008–2017. Source: Author's calculations on China's Year Book.

### 3.3.2. Coordination Characteristics

According to Equation (2), the coordination level between regional technological and institutional innovation capacity is determined. Figure 5 presents the average level of coordination between technological and institutional capacity in different regions of China over the period 2008–2017. It suggests that there are significant regional differences on the average coordination level of innovation capacity. As shown, the eastern curve and central curve almost have similar trend, which experience a dramatic fall in 2008, then go through a low but relatively stable stage of development from 2009 to 2012, while growing rapidly in 2012 (central curve) and 2014 (eastern curve), which peak at 2013 and 2015, respectively. Although the trend of coordinated development in the central and eastern regions is rather tortuous, it is relatively high during the sample period. While the level of coordination in the western region is stable, it is at a relatively low level as a whole. This confirms a fact that developed provinces like eastern and central regions of China are likely to pay more attention to the coordination of technological development and institutional development in green growth. Additionally, the higher coordination between technological innovation capability and institutional innovation capability in early stage may be due to the lower level of both, which leads to the higher level of similarity. Since then, the trend from low-level coordination to high-level coordination reflects the development trend of China's innovation policy.

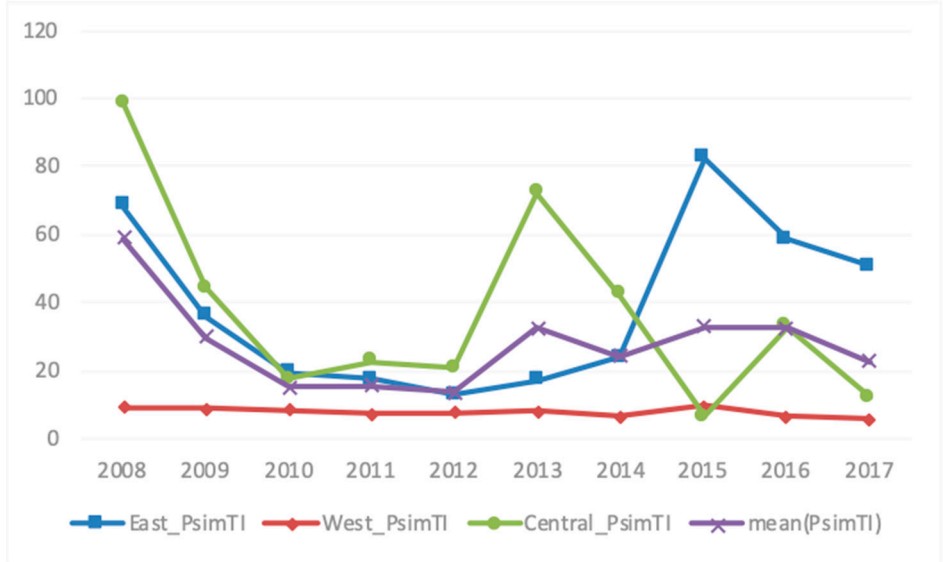

**Figure 5.** The average level of coordination between regional innovation capacity from 2008–2017. Source: Author's calculations on China's Year Book.

After the 2008 financial crisis, long-term government investment has weakened the independent innovation ability of Chinese enterprises to a certain extent, and consequently, economic growth depends on large-scale investment. In order to change the traditional mode of economic development, the report of the Eighteenth National Congress of the Communist Party of China in 2012 puts forward the implementation of "innovation-driven development strategy", emphasizing the important role of technological innovation in economic growth. In 2016, the Outline of National Innovation-Driven Development Strategy puts forward that innovation-driven economy is a systematic process, which highlights the two-wheel drive of "technological innovation and institutional innovation" mechanism at the national level.

## 4. Empirical Strategy

### 4.1. Model Specification

Based on the present literature on the drivers of green growth, this paper specifically addresses the causal effect of regional innovation capacity and its inter-coordination effect on the efficiency of green growth in different regions of China. Considering the complex relationship and dynamic effect between RIC and green growth efficiency, a baseline model is constructed as follows:

$$lnA_{it} = \beta_0 + \beta_1 Push_{it} + \beta_2 Coor_{it} + \beta_3 X'_{it} + \psi_{year} + \psi_{province} + \varepsilon_{it} \tag{3}$$

where $lnA_{it}$ indicates the green growth performance measured in each province of China, $i = 1$, ..., 30 indexes provinces in China, $t = 2008, \ldots, 2017$ indexes time. $Push_{it}$ indicates three specific groups of regional innovation capacity, including unified support, technical support, and systemic instruments; $Coor_{it}$ indicates the coordination level of technological innovation and institutional innovation; $X'_{it}$ indicates a group of control variables. Furthermore, fixed effects are introduced to capture non-observable province-specific and year-specific heterogeneity, $\psi_{province}$ and $\psi_{year}$ respectively. All the residual variation is captured by the error term $\varepsilon_{it}$.

Since green growth performance has a double truncation variable between 0 and 1, the estimation results of traditional OLS, GLS and GMM will be biased. Thus, the Tobit model is useful this paper to test the casual effect of RIC and its coordination effect on green growth performance. According to Li and Wang (2014) the linear econometric model is constructed as follows:

$$lnA_{it}^* = \beta_0 + \beta_1 Push_{it} + \beta_2 Coor_{it} + \beta_3 X_{it}' + \psi_{year} + \psi_{province} + \varepsilon_{it}$$
$$s.t.$$
$$A_{it}^* = A_{it}, \; if \; A_{it} \in [0,1]$$
$$A_{it}^* = 0, if \; A_{it} \in (-\infty, 0) \tag{4}$$
$$A_{it}^* = 1, if \; A_{it} \in (1, +\infty)$$

where $A_{it}$ represents the real green growth performance and $A_{it}^*$ indicates the potential counterpart; $\beta_1$ and $\beta_2$ reflects the effect of regional innovation capacity have on green growth.

### 4.2. Green Growth Measurement

Faced with the pressure of resources and environment and unbalanced regional development, the improvement of green growth efficiency has become a new criterion for judging economic efficiency in China. Based on this criterion, according to Lin and Liu (2015) [60], this paper constructs an econometric assessment model of China's regional green economic growth performance and takes the corresponding non-radial direction distance as the objective function.

Supposed that there are $K$ decision-making units (DMUs) and that energy ($E$) represents natural resource input; capital ($K$), and labor ($L$) are non-natural resource inputs; $Y^g$ represents the desirable economic output, and $Y^b$ are undesirable outputs that represent environmental pollutant discharge, respectively. According to Zhang et al. (2014) [61], the data envelope model of green development efficiency can be expressed as follows:

$$\vec{D}(K, L, E, Y^g, Y^b; g|CRS) = p^*$$
$$= \max \left\{ w^K \left( \sum_{n=1}^{N} \omega_n^K \alpha_n \right) + w^L \left( \sum_{n=1}^{N} \omega_n^L \beta_n \right) + w^E \left( \sum_{n=1}^{N} \omega_n^E \gamma_n \right) + w^g \left( \sum_{n=1}^{N} \omega_n^g \mu_n \right) + w^b \left( \sum_{n=1}^{N} \omega_n^b v_n \right) \right\}$$
$$s.t.$$
$$\sum_{n=1}^{N} z_n K_n \le K - \alpha_n g_n^K, \; \sum_{n=1}^{N} z_n L_n \le L - \beta_n g_n^L, \; \sum_{n=1}^{N} z_n E_n \le E - \gamma_n g_n^E \tag{5}$$
$$\sum_{n=1}^{N} z_n Y_n^g \le Y^g + \mu_n g_n^g, \; \sum_{n=1}^{N} z_n Y_n^b \le Y^b + v_n g_n^b$$
$$z_n \ge 0, n = 1, 2, \ldots, N \& \alpha, \beta, \gamma, \mu, v \ge 0$$

where the $\vec{D}(K, L, E, Y^g, Y^b; g|CRS) = p^*$ is a directional distance function, which measures the distance between the decision making unit and the effective production frontier; CRS means this model is under constant returns to scale; $g = (g^K, g^L, g^E, g^g, g^b)$ is the explicit directional vectors of natural resource input, other input, economic output, and environmental pollutant emission, which need to be set beforehand according to the objective function; $w = (w^K, w^L, w^E, w^g, w^b)$ denotes the corresponding index weight; $\alpha, \beta, \gamma, \mu, v$ represent the corresponding level of individual inefficiency measures for potential input/output, respectively; $K, L, E, g, b$ denote the directional vectors of $g$.

An indicator to measure unified performance in the context of the basic requirements of green growth requirements can be proposed. In order to pursue the maximization of $(\alpha, \beta, \gamma, \mu, v)$ at the same time, that is, economic output growth, resource conservation, and environmental pollutant reduction, the weight vector is set as $w^K = w^L = w^E = 1/9, w^g = w^b = 1/3$ and the directional vectors as $g = (-K, L, E, g, -b)$, based on Zhou et al. (2012) [62] and Barros et al. (2013) [63]. At this time, the total factor green development efficiency (*TDF*) is defined as follows:

$$TDF = \frac{1}{4} \left[ \frac{(1 - \alpha^*) + (1 - \beta^*) + (1 - \gamma^*) + (1 - v^*)}{1 + \mu^*} \right] \tag{6}$$

where $\alpha^*, \beta^*, \gamma^*, v^*, \mu^*$ and $\mu^*$ are the optimal solution of Model (1); $1/(1 + \mu^*), (1 - \alpha^*), (1 - \beta^*), (1 - \gamma^*)$ and $(1 - v^*)$ represent economic output efficiency, non-nature input efficiency, nature resource input

efficiency, and environmental emission efficiency, respectively, while the green development efficiency (*TDF*) is the comprehensive performance of five kinds of efficiency.

The indicators involved in the calculation process are as follows: human capital input (*L*) is replaced by the employment figures at the end of the year; energy input (*E*) is measured by the total energy consumption after converting into standard coal; capital stock in material capital input (*K*) is measured based on the method of perpetual inventory, which is proposed by Hu and Kao [64]. The calculation method is $K_t = (1 − \delta) K_{t−1} + I_t$, in which $K_t$ is the capital stock of the period *t*; it denotes the investment of the period t is replaced by the total amount of fixed capital formation; $K_{t−1}$ represents the capital stock of the period *t−1*; $\delta$ is the depreciation rate, which is 5%; the desirable output ($Y^g$) is measured by the regional GDP; the non-desirable output ($Y^b$) are industrial waste water accounts for GDP, industrial waste gas accounts for GDP, and industrial solid waste accounts for GDP respectively.

### 4.3. Control Variables and Data Source

In addition to the control of individual and time fixed effects, three alternative variables are introduced as driving force in the knowledge production process in the estimation. Detailed information of the main variables is shown in Table 2.

### 4.3.1. Government Expenditure Scale

Measured by the ratio of fiscal expenditure to GDP of each province. This variable is used to find out whether the intervention of local government in economic development will lead to the improvement of economic efficiency.

### 4.3.2. Regional Environment Regulation Intensity

The regional environment regulation intensity is measured by the proportion of the total internal expenditure on environment protection activities in the regional public expenditure. Generally, the more investment in environment protection, the more effective the green technological innovation in the industry, which consequently leads to the reduction of pollutant emissions.

### 4.3.3. Industrial Structure

Present literature has empirically found that the situation of industrial structure has a great impact on their green growth performance. In this paper, the industrial structure is measured by the proportion of tertiary industry output value to total industrial output value.

This study uses China's province-level data on green growth efficiency and regional innovation capacity from a variety of sources. The data span the years 2008 to 2017. All the data are obtained from the China Environmental Statistics Yearbook (2008–2017), the China Statistical Yearbook (2008–2017), the China Technology Statistical Yearbook (2008–2017), and provincial statistical yearbooks and bulletins over the years, as well as the national research network statistical database and the Baiteng network statistical database. The data on the environmental protection expenditure come from the China Environmental Statistics Yearbook (2008–2017).

**Table 2.** Descriptive statistics for variables.

| Type | Variables | Context | Mean | Std. Dev | Min | Max |
|------|-----------|---------|------|----------|-----|-----|
| Dependent variable | | | | | | |
| | TDF | Green growth performance | 0.691088 | 0.144798 | 0.340514 | 1 |
| Independent variables | | | | | | |
| | Unified | Regional unified innovation capacity | 0.217823 | 0.127600 | 0.064334 | 0.751720 |
| | Tech | Technical innovation capacity | 0.210591 | 0.153973 | 0.03128 | 0.814081 |
| | Soft | Institutional innovation capacity | 0.227598 | 0.126689 | 0.081746 | 0.710404 |
| | Psim_TI | Coordination of Technology and Innovation | 30.08369 | 101.3656 | 1.999555 | 1288.439 |
| Control variables | | | | | | |
| | Scale | Governmental expenditure scale | 0.232165 | 0.098841 | 0.087045 | 0.626863 |
| | Regul | Environmental protection expenditure | 0.007338 | 0.004979 | 0.001279 | 0.036143 |
| | IND | Industrial structure | 46.36573 | 8.31467 | 19.014 | 61.5 |

Source: Author's calculations.

## 5. Empirical Results and Discussion

### 5.1. Benchmark Results: The Effect of RIC on Green Growth Performance

The Hausman test shows that the fixed-effect model is more effective for the baseline models (Equations (3) and (4)) than the random-effect model. First, a regression analysis is conducted to examine the the role played by regional innovation capacity and its internal coordination in green growth performance, which provide us with specific characteristics of the causal effect between RIC and green growth performance in each province. The relationship between regional innovation capacity and TDF is represented in Columns (1) to (4).

Table 3 shows all variables that relating to regional innovation capacity result positive and statistically effects on green growth performance, which suggests that regional innovation capabilities play a significant role in promoting the green growth under scrutiny, with elasticity at about 0.432. The results are consistent with H3a, suggesting that a series of innovation and sustainability-related strategies in China have recently achieved its initial success.

As for the sub RIC instruments, Table 3 suggest that regional technological capacity instruments are able to stimulate TDF, with elasticity at about 0.310. Institutional instruments also positively affect the TDF, with elasticity at about 0.233. Though the coefficients of Tech in TDF seem to be larger than those of Soft in TDF. The results above support H1 and H2, which confirms the fact that the role of RIC in driving green growth is reliable and consequently provides effective verification of our research hypotheses. While the coefficient of technology is larger than that of institution, which also reflects the fact that China's institutional innovation lags behind technological innovation. In the process of development, more attention is attached by local government to the construction of technology rather than institution and culture, especially in inland areas of China [30].

In terms of the coordination level of regional technological innovation and institutional innovation (Equation (2)), Table 3 also reports that there is a significant but negative effect of coordination on TDF after introducing the coordination characteristics items from Column (4), which suggests the fact that China's institutional construction and technological development is far from coordinated. Moreover, this backward innovation system is likely to damage the green economy. This may be due to the lack of intellectual property protection, enterprise credit, market financing, and public education investment in China [65,66]. Compared with developed countries, China's market system has great room for improvement.

**Table 3.** The effect of regional innovation capacity on green growth performance.

| | (1) | (2) | (3) | (4) |
|---|---|---|---|---|
| | **Unified** | **Tech** | **Soft** | **Tech & Soft** |
| TDF | | | | |
| Unified | 0.432 *** | | | |
| | (3.61) | | | |
| Tech | | 0.310 *** | | 0.319 *** |
| | | (3.15) | | (3.51) |
| Soft | | | 0.233 ** | 0.197 ** |
| | | | (2.33) | (2.13) |
| Psim_IT | | | | −0.0032 *** |
| | | | | (−3.46) |
| Scale | 0.315 * | 0.377 ** | 0.293 * | 0.333 ** |
| | (1.85) | (2.24) | (1.63) | (2.05) |
| Regul | −6.298 ** | −6.411 ** | −6.008 * | −6.031 ** |
| | (−2.08) | (−2.11) | (−1.95) | (−2.13) |
| IND | −0.0385 | −0.0372 | −0.0503 * | −0.0317 |
| | (−1.38) | (−1.31) | (−1.80) | (−1.10) |
| year | yes | yes | yes | yes |
| province | yes | yes | yes | yes |
| _cons | 0.589 *** | 0.601 *** | 0.635 *** | 0.600 *** |
| | (11.32) | (11.51) | (12.50) | (10.66) |
| N | 290 | 290 | 290 | 290 |
| AIC | −497.8 | −495.1 | −491.3 | −494.6 |
| BIC | −468.4 | −465.8 | −462.0 | −457.9 |

t statistics in parentheses. * $p < 0.1$, ** $p < 0.05$, *** $p < 0.01$.

## 5.2. Regional Differences

Although the above results reveal the impact of regional innovation capacity on the overall green growth performance in China, the characteristics of regional differences are not considered. Thus, this section carries out further empirical research in eastern areas and central and western regions to characterize the different effects of the RIC instruments on green growth performance in different regions.

Results are outlined in Table 4. We test the causal effect of regional innovation capacity as well as the coordination characteristics on green growth in eastern region (Columns (1), (4), (7) and (10)), western region (Columns (2), (5), (8) and (11)) and central region (Columns (3), (6), (9) and (12)) respectively. In Columns (1) to (3) we use the unified innovation capacity as regional innovation capacity instruments. It shows that the unified capacity instrument is significant in both eastern region and central region, with 0.476 and 0.471, respectively. This is consistent with the relatively high level of economic development in the eastern and central regions.

Columns (4) to (6) use technology-related instruments to capture the relationship between regional technical innovation capacity and green growth performance. The coefficients show positive and statistical effect for all regions, with 0.293, 0.497, and 0.365, respectively, suggesting that technical innovation capabilities play a remarkable role in promoting green growth. It further confirms the regional technological innovation construction work in the green growth process of China. While the western region is still in its initial developmental stage, the lower energy consumption and the lower pollution emission result in a relatively higher green growth performance [67].

**Table 4.** The effect of RIC and its coordination on TDF.

| | (1) | (2) | (3) | (4) | (5) | (6) | (7) | (8) | (9) | (10) | (11) | (12) |
|---|---|---|---|---|---|---|---|---|---|---|---|---|
| | East_U | West_U | Cental_U | East_T | West_T | Cental_T | East_I | West_I | Cental_I | East_TI | West_TI | Cental_TI |
| TDF | | | | | | | | | | | | |
| Unified | 0.476 ** | 0.450 | 0.471 ** | | | | | | | | | |
| | (2.34) | (1.43) | (2.21) | | | | | | | | | |
| Tech | | | | 0.293 * | 0.497 ** | 0.365 ** | | | | 0.147 | 0.522 ** | 0.314 ** |
| | | | | (1.74) | (2.30) | (2.33) | | | | (0.87) | (2.22) | (2.14) |
| Soft | | | | | | | 0.458 ** | −0.0725 | 0.202 | 0.423 ** | −0.0388 | 0.154 |
| | | | | | | | (2.48) | (−0.39) | (1.49) | (2.38) | (−0.21) | (1.27) |
| Psim_TI | | | | | | | | | | 0.000172 * | −0.000807 | −0.000109 ** |
| | | | | | | | | | | (1.75) | (−0.62) | (−2.75) |
| Scale | −0.186 | 0.796 *** | −0.301 | −0.349 | 0.779 *** | −0.341 | −0.396 | 0.922 *** | −0.281 | −0.0671 | 0.672 *** | −0.202 |
| | (−0.24) | (3.44) | (−0.97) | (−0.45) | (3.60) | (−1.09) | (−0.55) | (4.21) | (−0.86) | (−0.13) | (3.02) | (−0.62) |
| Regul | −2.575 | −6.693 * | −7.992 | −5.812 | −6.655 * | −7.880 | 0.862 | −7.330 ** | −7.978 | −0.957 | −6.565 * | −1.535 |
| | (−0.34) | (−1.85) | (−1.44) | (−0.75) | (−1.89) | (−1.43) | (0.11) | (−2.04) | (−1.38) | (−0.12) | (−1.94) | (−0.28) |
| IND | −0.00204 | −0.000238 | −0.00260 * | −0.00451 | 0.000884 | −0.00253 * | −0.00208 | 0.000022 | −0.00272 ** | −0.000810 | 0.00120 | −0.00235 * |
| | (−0.44) | (−0.10) | (−1.96) | (−1.02) | (0.36) | (−1.91) | (−0.46) | (0.01) | (−2.00) | (−0.17) | (0.48) | (−1.81) |
| year | yes | yes | yes | yes | yes | yes | yes | yes | yes | yes | yes | yes |
| province | yes | yes | yes | yes | yes | yes | yes | yes | yes | yes | yes | yes |
| _cons | 0.701 ** | 0.386 ** | 0.869 *** | 0.884 *** | 0.361 ** | 0.887 *** | 0.731 ** | 0.448 *** | 0.896 *** | 0.620 * | 0.410 *** | 0.967 *** |
| | (2.11) | (2.51) | (7.18) | (2.82) | (2.46) | (7.45) | (2.32) | (2.85) | (6.98) | (1.85) | (2.61) | (7.84) |
| N | 110 | 90 | 100 | 110 | 90 | 100 | 110 | 90 | 100 | 110 | 90 | 100 |
| AIC | −123.5 | −177.8 | −253.3 | −121.1 | −181.0 | −253.8 | −124.2 | −175.9 | −250.8 | −121.1 | −177.8 | −254.6 |
| BIC | −101.9 | −157.8 | −232.4 | −99.54 | −161.0 | −232.9 | −102.6 | −155.9 | −229.9 | −94.08 | −152.8 | −228.5 |

t statistics in parentheses * $p < 0.1$, ** $p < 0.05$, *** $p < 0.01$.

Columns (7) to (9) use institutional instruments to test the effect of RIC on green growth performance. As the coefficient demonstrates, the effect of institutional instruments on TDF is positive and significant only in the eastern region (0.458), while there is no significant correlation in the western and central regions, and the coefficient in the western region is negative. Institutional innovation capacity only effectively promotes green growth in the eastern region, which is consistent with the fact that the central and western regions unilaterally pursue high returns in the process of development, ignoring the adjustment and improvement of institutional arrangements [30].

Columns (10) to (12) use coordination level of technical and institutional innovation as RIC characteristics to test the effect on green growth. There is positive and significant effect on TDF in eastern regions, while negative and statistic effect on central region and no significance on western region, which suggests that there are obvious regional differences in the effect of coordination between technological innovation and institutional innovation on green growth performance.

To sum up, there is still much room for promoting RIC nowadays in China: Firstly, in areas with high economic development, their industrial base is often relatively better, which will inevitably lead to increased environmental pollution to a certain extent; in addition, one-sided pursuit of economic benefits and neglect of ecological environment construction are the problems existing in the development of most cities in China, which consequently results in poor green growth performance. Secondly, in the process of undertaking industrial transfer, the central region undertakes a large number of industries with high energy consumption and pollution, which leads to increasingly serious environmental problems. Thirdly, despite the low level of economic development in the western region, energy input and pollution emissions in the economic process are far lower than those in the central and eastern regions, which makes the overall innovation driving effect in the western region more significant.

It suggests that the regional innovation capacity is positive and significant to induce green growth, while there are great differences among the three regions of China. This may be due to the high level of economic development in the eastern region, the comparative advantages of human capital and technological basis, and the more rational institutional system. Therefore, variations of regional innovation capacity incentivize the green growth of the eastern region more significantly, while the central region has undertaken a large number of heavy pollution and high energy consumption enterprises and excessively pursued economic benefits in the process of economic development. Ignoring the ecological and social development, the regional technological innovation system is far from supporting the sustainable development of economy and society. While the western region is rich in natural resources, the level of economic development is low, and the industrial base is poor. Therefore, the pollution emissions are far lower than those in the central region, which makes the effect of regional innovation capacity in the western area slightly higher than that in the central region.

*5.3. Robustness Test*

The robustness tests of the estimation results are conducted to ensure more robust empirical results as follows:

5.3.1. Substitution of the Dependent Variable

Here we use the energy-environmental performance as a substitution of green growth performance, Following Zhang and Choi (2014) [46], we set the directional vector as $g = \left(0, 0, -g^E, g^g, -g^b\right)$, and the weight vector as $w^E = w^g = w^b = 1/3$ and $w^K = w^L = 0$ to remove the effect of capital and labor, which may not contribute to emissions directly. Then the energy-environmental related green growth performance (*EDF*) is defined as follows:

$$EDF = \frac{1}{2}\left[\frac{(1 - \gamma^*) + (1 - v^*)}{1 + \mu^*}\right] \tag{7}$$

where $\gamma^*, \upsilon^*$ and $\mu^*$ are the optimal solution of model 1; $1/(1+\mu^*), (1-\gamma^*)$ and $(1-\upsilon^*)$ represent economic output efficiency, nature resource input efficiency, and environmental emission efficiency respectively, while the energy-environmental performance (*EDF*) is the comprehensive performance of three kinds of efficiency.

### 5.3.2. Endogenous Test

The Hausman test also indicates that there are endogenous problems. Instrumental variables are often used to tackle these problems. These variables are highly correlated with endogenous explanatory variables but not with residual variables. Strictly exogenous instrumental variables are often difficult to find, so the lagged term of explanatory variables is frequently used in this process. Thus, the lagged value of policy is used as an instrumental variable in this paper through a two-stage least squares method (2SLS). The results are shown in Table 5.

The results are basically consistent with the benchmark model from Table 3. There is no substantial change of the coefficients or significance from Columns (1) to (8). Although the significance differs in institutional instruments (Columns (3) and (7)), the signs of the estimated coefficients are the same as in Table 3. In addition, the test statistics in each column of Table 5 show that the corresponding *p* value of the Sargan test is greater than 0.05, and the original hypothesis cannot be rejected, which explains that the instrumental variables applied in the model are effective and satisfy the exogenous conditions.

**Table 5.** Robust test.

| | (1) | (2) | (3) | (4) | (5) | (6) | (7) | (8) |
|---|---|---|---|---|---|---|---|---|
| | U_EDF | T_EDF | I_EDF | TI_EDF | U_2SLS | T_2SLS | I_2SLS | TI_2SLS |
| main | | | | | | | | |
| Unified | 0.898 *** | | | | 0.593 *** | | | |
| | (6.58) | | | | (3.48) | | | |
| Tech | | 0.753 *** | | 0.740 *** | | 0.405 *** | | 0.413 * |
| | | (7.13) | | (7.24) | | (2.61) | | (1.65) |
| Soft | | | 0.0982 | 0.0683 | | | 0.737 *** | 0.216 |
| | | | (0.85) | (0.68) | | | (4.00) | (0.66) |
| Psim_TI | | | | −0.00027 ** | | | | −0.0003 *** |
| | | | | (−2.88) | | | | (−3.13) |
| Scale | −0.507 *** | −0.425 ** | −0.325 * | −0.5363 ** | −0.0282 | 0.0756 | −0.363 * | 0.0455 |
| | (−2.85) | (−2.42) | (−1.59) | (−3.10) | (−0.15) | (0.39) | (−1.74) | (0.17) |
| Regul | −4.313 | −5.032 * | −4.982 | −5.572 * | −1.928 | −2.185 | −1.437 | −1.748 |
| | (−1.39) | (−1.64) | (−1.51) | (−1.88) | (−0.61) | (−0.69) | (−0.44) | (−0.59) |
| IND | 0.0215 | 0.0353 | 0.00437 | 0.0364 | −0.0161 | −0.0135 | 0.0268 | −0.0289 |
| | (0.75) | (1.22) | (0.14) | (1.24) | (−0.54) | (−0.43) | (0.78) | (−0.71) |
| year | yes | yes | yes | yes | yes | yes | yes | yes |
| province | yes | yes | yes | yes | yes | yes | yes | yes |
| _cons | 0.364 *** | 0.377 *** | 0.502 *** | 0.377 *** | 0.593 *** | 0.608 *** | 0.603 *** | 0.683 *** |
| | (6.38) | (6.58) | (7.67) | (6.10) | (9.24) | (8.73) | (11.78) | (5.63) |
| N | 290 | 290 | 290 | 290 | 232 | 232 | 232 | 232 |
| AIC | −487.0 | −495.3 | −453.6 | −491.9 | | | | |
| BIC | −457.6 | −465.9 | −424.3 | −455.2 | | | | |
| Sargan test | | | | | 1.512 | 0.899 | 6.512 | 2.820 |
| *p*-value | | | | | 0.2188 | 0.3429 | 0.0107 | 0.2441 |

t statistics in parentheses. * $p < 0.1$, ** $p < 0.05$, *** $p < 0.01$.

## 6. Conclusions and Policy Implications

This paper examines the effects of regional innovation capacity and its internal characteristics on green growth in a cross-section of China's provinces over the period 2008–2016. We creatively construct an evaluation system that captures regional innovation capacity and the coordination effect of its sub-items influencing the dynamics of green growth in China. Moreover, our paper originally constructs the econometric assessment of this issue in the case of China's regional innovation capacity, while there is still much room to improve on it in future studies when more refined, quantitative information becomes available.

According to previous empirical literature on the green economic domains, most studies suggest that green growth is driven by technological instruments [24]. In addition, our analysis gives an insight into the role played by technological and institutional innovation instruments as well as their coordination level, which result positive while statistically weakly significant in the eastern region of China during the sample period. According to the general idea that the coordination of regional innovation systems across different sectors may strengthen the ability of RIC to promote green growth, we also analyze the role of unified RIC instruments in the development of regional green growth. Specifically, our analysis focuses on the causal effects of these three selected instruments of the RIC on green economy performance in each province, namely, technological innovation capacity, institutional and systemic capacity, and the unified one. In doing so, we investigate the role of specific efforts that each province has on shaping regional innovation capacity. With the data constraints, the provided empirical results reveal that technology-related instruments play a more relatively significant role than institutional instruments on the development of green growth in most regions of China. Moreover, by examining the coordination effect of technical and institutional instruments on green growth performance. Our analysis also suggests that the institutional construction lags far behind technological development in China.

Our findings also have important policy implications. The empirical results illustrate that RIC has a very positive influence on the development of the green economy in China, while the regional difference might harm the approach. The complicated relationship between innovation system and green growth shows that many factors should be considered in the development of relevant policies, such as environmental problems, economic problems, and regional characteristics. Thus, the public policy support for innovation should vary along with different times and locations according to the actual situation of different provinces.

Moreover, there is still considerable scope for China's government to develop both regional innovation capacity (especially the institutional innovation system) and green technology. China should further expand the scale and intensity of government subsidies in the field of environmental protection and encourage enterprises to invest in environmental protection. The eastern part of China has the highest degree of economic development but because of its lack of independent innovation ability, poor market mechanisms, and serious brain drain, it has encountered obstacles to industrial transformation. This consequently results in inertia in the process of green technical innovation and damages the manufacturing industry, thus restricting the improvement of China's economic level as well. As for the central and western parts, more governmental support of green technologies is needed to induce green innovation.

While the results are interesting and robust, further work on this issue could be undertaken. This includes accounting for variations in the actual environment of China, investigating the determinants of green technologies, and conducting a better examination of dynamic issues.

**Author Contributions:** Conceptualization, S.H. and S.L.; methodology, S.H.; software, S.H.; validation, S.H., D.L., and S.L.; formal analysis, D.L.; resources, S.H.; data curation, S.H.; writing—original draft preparation, S.H.; writing—review and editing, D.L.; supervision, S.L. All authors contributed to writing the paper.

**Funding:** The authors are grateful to the financial support provided by the National Natural Science Foundation of China (71774128), the National Social Science Foundation of China (19BJY057).

**Acknowledgments:** We are grateful to acknowledge support from Qian Yu, Chi Zhang, and Haoqiang Wu for their comments on earlier drafts of this paper, as well as seminar participants at Wuhan University of Technology during the initial writing of this paper. We owe special thanks to Kailey Shi, Rongrong Chen, Koko Qian, Qianlin Du, and Tingyu Lee for their assisting in data collection.

**Conflicts of Interest:** The authors declare no conflict of interest.

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
