# Peer review of "How Does Regional Innovation Capacity Affect the Green Growth Performance? Empirical Evidence from China"

_sustainability, doi:10.3390/su11185084_

Round 1

Reviewer 1 Report

The focus of the paper seems very interesting and the empirical part seems to be solidly constructed. However, we believe that the paper can improve if the following observations are taken into account:

1) In line 19 (in the “Abstract) it is indicated:

“(i) RIC significantly affect the green growth performance of 30 provinces in China, showing regional differences.”

In the "Abstract" should not appear acronyms whose meaning is not properly detailed. Not all the public interested in the article has to know these acronyms. Therefore, in the same “Abstract” it should be noted above what is the meaning of RIC. This previous definition can be done on line 15:

“significance to assess the effect of regional innovation capacity (RIC) on the green growth performance.”

2) On line 55 the call of footnote 2 must be indicated by superscript.

3) Sometimes the quotes in the paper do not follow the Sustainability style. For example:

Lines 145-146: “Wu et al. (2018) shows that the roles and positions of the universities…”

Line 148: “Using China’s provincial data, Liu et al. (2019) shows that…”

This happens in more parts of the paper. We believe that this must be corrected.

4) The following quote appears on line 200: “Liu (2019) [8]”.

However, in the references list, appointment 8 does not correspond with Liu but with Stahel. In addition, in the list of references we have not found the reference Liu (2019).

5) In our humble opinion, the "Summary", "Introduction" and "Literature review and research hypothesis" sections should be rewritten, as they exhibit overt linguistic inconsistencies. Let's see some examples:

Lines 24-26: “(iii) the coordination of technical and institutional instruments has a significant but negative effect on green growth performance. While positive in the eastern region and negative in central region respectively.”

It cannot be stated categorically that “the coordination of technical and institutional instruments has a significant but negative effect on green growth performance”, since the results indicate that the relationship is contradictory. In fact, the contradiction is recognized in the following paragraph: "While positive in the eastern region and negative in central region respectively".

Lines 159-161: “Although the instruments mentioned above seem to be related to the increase in green innovation activities. They are frequently implemented simultaneously, it is necessary to constitute such a mix of instruments to increase inventive activity, especially for green/clean innovation.”

It is obvious that there is no consistency in the drafting of the two preceding paragraphs, since the beginning of the sentence (Although) announces something that a period after the word "activities" truncates.

Of this tenor there are more inconsistencies in the text

6) In our opinion, the theoretical and empirical foundations on which the three hypothesis approach is based is too poor.

7) Finally, underline that in the section “Empirical results and Discussion” there is a presentation of the empirical results obtained, but not a true discussion of them, since at no time are these results confronted with previous theoretical and empirical research (Chinese and international ).

I hope my comments will help you improve your paper.

Good luck.

Author Response

 Thank you for your letter and for the reviewers’ comments concerning our manuscript entitled “How does Regional Innovation Capacity Affect the Green Growth Performance? Empirical Evidence from China ” (ID: sustainability-563728). Those comments are all valuable and very helpful for revising and improving our paper, as well as the important guiding significance to our researches. We have studied comments carefully and have made correction which we hope meet with approval. Revised portion are marked in red in the paper. The main corrections in the paper and the responds to the reviewer’s comments are as flowing:
Responds to the reviewer’s comments:
Reviewer #1: 

1. Response to comment: We agreed with that “In the "Abstract" should not appear acronyms whose meaning is not properly detailed.” Considering the this suggestion, we have re-written the abstract as below:

Actions: the statements of line 15 “ This paper firstly exploits a model to measure regional innovation capacity from the perspective of technological and institutional respect.” are corrected as “ It is of great practical significance to assess the effect of regional innovation capacity (RIC) on the green growth performance. ”

And line 19“(i) RIC significantly affect the green growth performance of 30 provinces in China, showing regional differences.” are corrected as “(i) regional innovation capacity significantly affect the green growth performance of 30 provinces in China, showing regional differences.”

2. Response to comment: We agree with that all the footnote must be indicated by superscript.

Actions: the statements of line 55 “However, according to a resent published report, China’s innovation index ranks 17th in the world in 2018, and 70th in the sub-item of institutional environment2” are corrected as “However, according to a resent published report, China’s innovation index ranks 17th in the world in 2018, and 70th in the sub-item of institutional environment2”

3. Response to comment: We are very sorry for our incorrect writing of the quotes in the paper, and have made correction according to the Reviewer’s comments as below:

Actions: the statements of line 145-146 “Wu et al. (2018) shows that the roles and positions of the universities…” are corrected as “Wu et al. Shows that the roles and positions of the universities...”;

the statements of line 148 “Using China’s provincial data, Liu et al. (2019) shows that…” are corrected as “Using China’s provincial data, Liu et al. shows that…”;

For all correction, please refer to the new manuscript (marked version)

4. Response to comment: It is really true as Reviewer suggested that appointment 8 does not correspond with Liu in the references list.

Actions: The reference citation are added in the references list.

5. Response to comment: As Reviewer suggested that the "Summary", "Introduction" and "Literature review and research hypothesis" sections should be rewritten, as they exhibit overt linguistic inconsistencies. Considering the this suggestion, we have re-written these part as below:

 Actions: the statements of lines 24-26: “(iii) the coordination of technical and institutional instruments has a significant but negative effect on green growth performance. While positive in the eastern region and negative in central region respectively.” are corrected as “(iii) the coordination of technical and institutional instruments has a significant effect on green growth performance, positive in the eastern region and negative in central region respectively. ”;

the statements of lines 159-161: “Although the instruments mentioned above seem to be related to the increase in green innovation activities. They are frequently implemented simultaneously, it is necessary to constitute such a mix of instruments to increase inventive activity, especially for green/clean innovation.”are corrected as “Although the instruments mentioned above seem to be related to the increase in green innovation activities, most of them focus on the distinct role of innovation instruments in shaping regional innovation capacity. While they are frequently implemented simultaneously in practice, it is necessary to constitute such a mix of instruments to increase inventive activity, especially for green/clean innovation.”

For all correction, please refer to the new manuscript (marked version)

6. Response to comment: It is really true as Reviewer suggested that the theoretical and empirical foundations on which the three hypothesis approach is based need to be improved. Considering the this suggestion, we have re-written these part as below:

Actions: 2.Literature Review and Research Hypotheses

The main objective of this paper is to test the effect of two types of regional innovation capacity (RIC) on green and sustainable economic development. Previous studies have explored various factors affecting green growth from different perspectives [26][27][28][29]. With the rise of national and regional innovation system theory in the 1990s, there is a growing attention of literature on the role of regional innovation capacity. The regional innovation capacity has been found to be of equal importance as the green growth drivers [30].

In the early work of Freeman[6] and Nelson[31], institutional structure and cultural differences in innovation are the key factors contributing to the national differences in innovation performance. Since then, RIC scholars have developed a series of theories and empirical strategies related to regional differences and socio-technical innovation from the perspective of earlier static analysis to current dynamic approach[4]. These studies highlight both technological advance and institutional structure as vital roles in the green transition processes, though the neoclassical economics [32] and new institutional economics [33] have contribute to finding out which forces drives green growth, and consequently adopts different emphasis on the roles of technology and institution in green growth respectively[14] [34].

On the technical sides, previous studies suggest that research and development (R&D) activities are very important innovation instruments in shaping regional technical innovation capacity[35][36]. While it is well-known that R&D activities are highly uncertain, indivisible and non-exclusive, which may lead to market failure and insufficient R&D investment, especially in basic research fields. Thus, the production and diffusion of knowledge and innovation is efficient only when government, research institutions and enterprises interact in a constructive, interactive and complementary way[37]. In this respect, public policies are likely to provide a favourable situation for technical innovation and capacities acquired by R&D activities. Therefore, R&D policy instruments are widely used in the broad analytic framework of RIC over the years[38][39]. With respect to green growth, public subsidy mechanism plays an vital role in inducing ecological innovation and improving regional innovation capacities, which consequently reduce the cost of pollution and emission abatement. The existing works of literature discussed the role of technological innovation in green growth mainly on specific technological fields, such as technological upgrading of new energy automotive industry, improvement of clean energy productivity [40][41][42]. These studies emphasize that technological innovation capacity is an effective means to promote regional green economy development in the long run [24][43][44].

With focus on the development of a regional green economy, these analyses are expanded by studying the impact of regional technological innovation capability and its coordination characteristics on green growth performance. Since technical progress is the basis for the green economic growth. It is expected that regional technological innovation capacity fosters inventive activity and thereby increases the performance of green growth.

The first hypothesis taken in this paper is stated as follows:

H1: Regional technological innovation capacity increases the performance of green growth.

With respect to the institution sides, a growing number of recent studies pay attention to the role played by institutional factors in fostering green/clean technology in the context of market failures and environmental externality [4][24] [45]. For example, the importance of the technology policy is highlighted in [28] [46] and the significance of the environmental policy is discussed in [47][48]. These studies highlight the primary role played by RIC in driving green growth. They also suggest that the perfection of the institutional mechanism is of great significance to innovation [1][4]. Acemogluet al. examines how institutional change affects the development of a country or region. It is found that the fundamental institutional changes brought about by the French Revolution led to the long-term economic growth in the occupied areas[49]. Kirchherr et al. points out that institutional structure and cultural difference are likely to be circular economy barriers rather than a single technological barrier[50].

The constructions of regional, institutional, and systemic innovation capacity aim to connect various inventive actors, such as enterprises, research colleges, and relevant institutes. Their inventive cooperation is fostered through learning, technology and resources sharing process among those parties [19][28]. Those constructions include public education and information support instruments, market system support instruments, financial support instruments and government policy support instruments. They help to shape green growth by providing infrastructure, building knowledge exchange and technology transfer platforms. Facilitating knowledge learning and exchange and to enhance cooperation is expected [29]. There is extensive empirical literature on the role of institutional instruments in shaping green growth. Wu et al. shows that the roles and positions of the universities are the most important in the cooperative of the industry-university-research institution, which is one of the main ways for enterprises to gain competitiveness [51]. Using China’s provincial data, Liu et al. shows that the perfection of the financial system has a significant effect on green economic growth. They pointed out that the positive role of the stock market is stronger than that of the banking sector. In other words, the market-oriented financial structure system is more conducive to the long-term growth of China's green economy [30]. These institutional instruments provide various public support for promoting the green economy. Furthermore, by providing various incentives and improving the institutional environment, it is possible to increase the performance of green growth by reducing the risk of innovation investment and forming cooperation with more potential partners. Thus, the second hypothesis in this paper is highlighted as follow:

H2: Regional institutional innovation capacity increases the performance of green growth.

Although the instruments mentioned above seem to be related to the increase in green innovation activities, most of them focus on the distinct role of innovation instruments in shaping regional innovation capacity. While they are frequently implemented simultaneously in practice, it is necessary to constitute such a mix of instruments to increase inventive activity, especially for green/clean innovation. Michael P. Todaro believes that economic development should be viewed as a multidimensional process including the restructuring and restructuring of the entire economic and social system [52]. However, a variety of empirical studies isolate technological approach and institutional approach in their analytic framework, and they did not explore how their interaction between them affects green growth. The intrinsic motive force of economic development depends on technological progress and the construction of corresponding institution structure, which is the process of coordination and unification of technology and institution, Furman further pointed out that the key to building an innovation-driven system is to maintain and coordinate the relationship between these factors[53].

Since the Reform and Development of China, there experienced continuous optimization of the institutional environment and the upgrading of technological level. Both technology and institutions are likely to contribute to China's green growth performance and total factor productivity (TFP). One of the key factors to the sustainability of technological innovation is whether the current institutional arrangements provide suitable conditions for the occurrence and diffusion of technological innovation. Another is whether there is an effective property rights system that stimulates innovation and reduces risks of innovation, including private property rights, patent systems and intellectual property protection systems, is the key to the sustainability of technological innovation.

Based on the existing research findings, it is suggested that both technological and institutional instruments should be included in measuring regional innovation capacity. In addition, the role of the coordination level between technological and institutional innovation in promoting green growth was tested as well. It is expected that both technology and institution create incentives.  Thus, the following hypothesis in this paper is highlighted:

H3a: The performance of green growth is significant and positively affected by the combination of technological instruments and institutional instruments.

H3b: The level of coordination between regional technological innovation capacity and institutional innovation capacity has a positive effect on green growth performance.

It is well-known that technological innovation directly affects economic growth. The effects are from the developing green and clean technology to realizing the recycling of resources, optimizing the energy consumption structure, and improving the utilization rate of resources. While institution and systemic innovation aims to connect different innovators, such as universities, scientific research institutions, and enterprises, and encourage knowledge and technology learning and resource sharing by building knowledge exchange and technology transfer platforms, it includes infrastructure construction and encourages various forms of technical cooperation among innovators [54][55]. The combination of these two types of innovation capacity constitutes the regional innovation system, which needs to be consistent to maintain the long-term sustainability of green growth.

7. Response to comment: It is really true as Reviewer suggested that the ““Empirical results and Discussion” section should confronted with previous theoretical and empirical research. Considering the this suggestion, we have re-written these part as below:

Actions: 5.Empirical results and Discussion

5.1.Benchmark results: the effect of RIC on green growth performance

The Hausman test shows that the fixed-effect model is more effective for the baseline models (equations (3) and (4)) than the random-effect model. First, a regression analysis is conducted to examine the the role played by regional innovation capacity and its internal coordination in green growth performance. Which provide us specific characteristics of the causal effect between RIC and green growth performance in each province. The relationship between regional innovation capacity and TDF is represented in column (1)-(4).

Table 3 shows all variables that related to regional innovation capacity result positive and statistically effect on green growth performance, which suggests that regional innovation capabilities play a significant role in promoting green growth under scrutiny, with elasticity at about 0.432. The results are consistent with H3a, suggesting that a series of innovation and sustainability-related strategies in China have recently achieved its initial success.

As for the sub RIC instruments, Table 3 suggest that regional technological capacity instruments are able to stimulate TDF, with elasticity at about 0.310. Institutional instruments also positively affect the TDF, with elasticity at about 0.233. Though the coefficients of Tech in TDF seems to be larger than that of Soft in TDF. The results above support H1 and H2 ,which confirm to the fact that the role of RIC in driving green growth is reliable and consequently provide effective verification of our research hypotheses. While the coefficient of technology is larger than that of institution, which also reflects the fact that China's institutional innovation lags behind technological innovation. In the process of development, more attention is attached by local government to the construction of technology rather than institution and culture, especially in inland areas of China [30].

In terms of the coordination level of regional technological innovation and institutional innovation (equation (2)), table 3 also reports that there is significant but negative effect of coordination on TDF after introducing the coordination characteristics items from column (4). Which suggest the fact that China's institutional construction and technological development is far from coordinated. And this backward innovation system is likely to damage the green economy. This may be due to the lack of intellectual property protection, enterprise credit, market financing, and public education investment in China[67][68]. Compared with developed countries, China's market system has great room for improvement.

5.2.Regional differences

Although the above results reveal the impact of regional innovation capacity on the overall green growth performance in China, while the characteristics of regional differences are not considered. Thus, this section carries out further empirical research in eastern areas, central and western regions to characterize the different effects of the RIC instruments on green growth performance in different regions.

Results are outlined in Table 4. We test the causal effect of regional innovation capacity as well as the coordination characteristics on green growth in eastern region (column (1),(4),(7) and (10)), western region (column (2),(5), (8) and (11)) and central region (column (3),(6), (9) and (12)) respectively. In columns (1) to (3) we use the unified innovation capacity as regional innovation capacity instruments. It shows that the unified capacity instrument is significant in both eastern region and central region, with 0.476 and 0.471 respectively. This is consistent with the relatively high level of economic development in the eastern and central regions.

Column (4) to (6) use technology-related instruments to capture the relationship between regional technical innovation capacity and green growth performance. The coefficients show positive and statistical effect for all regions, with 0.293, 0.497 and 0.365 respectively, suggesting that technical innovation capabilities play a remarkable role in promoting green growth. It further confirm that the regional technological innovation construction work in the green growth process of China. While the western region is still in its initial developmental stage, the lower energy consumption and the lower pollution emission, resulting in a relatively higher green growth performance[2][69].

Column (7) to (9) use institutional instruments to test the effect of RIC on green growth performance. As the coefficient demonstrates, the effect of institutional instruments on TDF is positive and significant only in eastern region (0.458). While there is no significant correlation in the western and central regions, and the coefficient in the western region is negative. Institutional innovation capacity only effectively promotes green growth in the eastern region, which is consistent with the fact that the central and western regions unilaterally pursue high returns in the process of development, ignoring the adjustment and improvement of institutional arrangements [30].

Column (10) to (12) uses coordination level of technical and institutional innovation as RIC characteristics to test the effect on green growth. While there is positive and significant effect on TDF in eastern regions, while negative and statistic effect on central region and no significance on western region. Which suggest that there are obvious regional differences in the effect of coordination between technological innovation and institutional innovation on green growth performance.

To sum up, there is still much room for promoting RIC in nowadays China: firstly, in areas with high economic development, their industrial base is often relatively better, which will inevitably lead to increased environmental pollution to a certain extent. In addition, one-sided pursuit of economic benefits and neglect of ecological environment construction are the problems existing in the development of most cities in China, Which consequently resulting in poor green growth performance; secondly, in the process of undertaking industrial transfer, the central region undertakes a large number of industries with high energy consumption and pollution, which leads to increasingly serious environmental problems; thirdly, despite the low level of economic development in the western region, energy input and pollution emissions in the economic process are far lower than those in the central and Eastern regions, which makes the overall innovation driving effect in the western region more significant.

Special thanks to you for your good comments.

Reviewer 2 Report

The article focused on the important topic that is under scope of the journal.

The authors analyze regional innovation capacity and green growth performance in China in 2008 – 2017.

The paper is clear and well-structured.

Analysis is done in the appropriate depth and range.

Methodology and goals are almost clearly presented.

Sufficient number of sources are used.

Results may be used in future.

Unfortunately, article has some limitations. The biggest is in the English and typing errors, for example:

row 31 – 33 – bad sentences row 37 – in the poor level1 – 1 = use upper index. The same is for all footnotes. row 46 – resent instead recent row 87 – the the etc.

Hypotheses have to be supported and discussed with the existing literature.

row 252 -  and a dramatic fall in 2016 – according to graph it was in 2015.

fig. 2 - 5 -  Author’s calculations – it is based on some data that are not cited.

row 506 – period is not correct

Author Response

Dear Editor and Reviewers:
Thank you for your letter and for the reviewers’ comments concerning our manuscript entitled “How does Regional Innovation Capacity Affect the Green Growth Performance? Empirical Evidence from China ” (ID: sustainability-563728). Those comments are all valuable and very helpful for revising and improving our paper, as well as the important guiding significance to our researches. We have studied comments carefully and have made correction which we hope meet with approval. Revised portion are marked in red in the paper. The main corrections in the paper and the responds to the reviewer’s comments are as flowing:
Responds to the reviewer’s comments:

Reviewer #2: 

1. Response to comment: We are very sorry for our poor English and typing errors in this paper, and have made correction according to the Reviewer’s comments as below:

Actions: the statements of lines 31-33 “Over the last two decades, characterized by high investment, high energy consumption and high emissions, China’s industrial GDP has achieved remarkable growth rate, with 11.5% annually” are replaced by “Green growth has attracted widespread attention over the last two decades, especially in China. It is believed that the extensive industrial growth model of China has already led to substantial consumption of resources and ecological deterioration of the environment. ”;

And the incorrect writing of word recent in row 46 and 87

2. Response to comment: It is really true as Reviewer suggested that the hypotheses have to be supported and discussed with the existing literature. Considering the this suggestion, we have re-written these part as below:

Actions: 2.Literature Review and Research Hypotheses

The main objective of this paper is to test the effect of two types of regional innovation capacity (RIC) on green and sustainable economic development. Previous studies have explored various factors affecting green growth from different perspectives [26][27][28][29]. With the rise of national and regional innovation system theory in the 1990s, there is a growing attention of literature on the role of regional innovation capacity. The regional innovation capacity has been found to be of equal importance as the green growth drivers [30].

In the early work of Freeman[6] and Nelson[31], institutional structure and cultural differences in innovation are the key factors contributing to the national differences in innovation performance. Since then, RIC scholars have developed a series of theories and empirical strategies related to regional differences and socio-technical innovation from the perspective of earlier static analysis to current dynamic approach[4]. These studies highlight both technological advance and institutional structure as vital roles in the green transition processes, though the neoclassical economics [32] and new institutional economics [33] have contribute to finding out which forces drives green growth, and consequently adopts different emphasis on the roles of technology and institution in green growth respectively[14] [34].

On the technical sides, previous studies suggest that research and development (R&D) activities are very important innovation instruments in shaping regional technical innovation capacity[35][36]. While it is well-known that R&D activities are highly uncertain, indivisible and non-exclusive, which may lead to market failure and insufficient R&D investment, especially in basic research fields. Thus, the production and diffusion of knowledge and innovation is efficient only when government, research institutions and enterprises interact in a constructive, interactive and complementary way[37]. In this respect, public policies are likely to provide a favourable situation for technical innovation and capacities acquired by R&D activities. Therefore, R&D policy instruments are widely used in the broad analytic framework of RIC over the years[38][39]. With respect to green growth, public subsidy mechanism plays an vital role in inducing ecological innovation and improving regional innovation capacities, which consequently reduce the cost of pollution and emission abatement. The existing works of literature discussed the role of technological innovation in green growth mainly on specific technological fields, such as technological upgrading of new energy automotive industry, improvement of clean energy productivity [40][41][42]. These studies emphasize that technological innovation capacity is an effective means to promote regional green economy development in the long run [24][43][44].

With focus on the development of a regional green economy, these analyses are expanded by studying the impact of regional technological innovation capability and its coordination characteristics on green growth performance. Since technical progress is the basis for the green economic growth. It is expected that regional technological innovation capacity fosters inventive activity and thereby increases the performance of green growth.

The first hypothesis taken in this paper is stated as follows:

H1: Regional technological innovation capacity increases the performance of green growth.

With respect to the institution sides, a growing number of recent studies pay attention to the role played by institutional factors in fostering green/clean technology in the context of market failures and environmental externality [4][24] [45]. For example, the importance of the technology policy is highlighted in [28] [46] and the significance of the environmental policy is discussed in [47][48]. These studies highlight the primary role played by RIC in driving green growth. They also suggest that the perfection of the institutional mechanism is of great significance to innovation [1][4]. Acemogluet al. examines how institutional change affects the development of a country or region. It is found that the fundamental institutional changes brought about by the French Revolution led to the long-term economic growth in the occupied areas[49]. Kirchherr et al. points out that institutional structure and cultural difference are likely to be circular economy barriers rather than a single technological barrier[50].

The constructions of regional, institutional, and systemic innovation capacity aim to connect various inventive actors, such as enterprises, research colleges, and relevant institutes. Their inventive cooperation is fostered through learning, technology and resources sharing process among those parties [19][28]. Those constructions include public education and information support instruments, market system support instruments, financial support instruments and government policy support instruments. They help to shape green growth by providing infrastructure, building knowledge exchange and technology transfer platforms. Facilitating knowledge learning and exchange and to enhance cooperation is expected [29]. There is extensive empirical literature on the role of institutional instruments in shaping green growth. Wu et al. shows that the roles and positions of the universities are the most important in the cooperative of the industry-university-research institution, which is one of the main ways for enterprises to gain competitiveness [51]. Using China’s provincial data, Liu et al. shows that the perfection of the financial system has a significant effect on green economic growth. They pointed out that the positive role of the stock market is stronger than that of the banking sector. In other words, the market-oriented financial structure system is more conducive to the long-term growth of China's green economy [30]. These institutional instruments provide various public support for promoting the green economy. Furthermore, by providing various incentives and improving the institutional environment, it is possible to increase the performance of green growth by reducing the risk of innovation investment and forming cooperation with more potential partners. Thus, the second hypothesis in this paper is highlighted as follow:

H2: Regional institutional innovation capacity increases the performance of green growth.

Although the instruments mentioned above seem to be related to the increase in green innovation activities, most of them focus on the distinct role of innovation instruments in shaping regional innovation capacity. While they are frequently implemented simultaneously in practice, it is necessary to constitute such a mix of instruments to increase inventive activity, especially for green/clean innovation. Michael P. Todaro believes that economic development should be viewed as a multidimensional process including the restructuring and restructuring of the entire economic and social system [52]. However, a variety of empirical studies isolate technological approach and institutional approach in their analytic framework, and they did not explore how their interaction between them affects green growth. The intrinsic motive force of economic development depends on technological progress and the construction of corresponding institution structure, which is the process of coordination and unification of technology and institution, Furman further pointed out that the key to building an innovation-driven system is to maintain and coordinate the relationship between these factors[53].

Since the Reform and Development of China, there experienced continuous optimization of the institutional environment and the upgrading of technological level. Both technology and institutions are likely to contribute to China's green growth performance and total factor productivity (TFP). One of the key factors to the sustainability of technological innovation is whether the current institutional arrangements provide suitable conditions for the occurrence and diffusion of technological innovation. Another is whether there is an effective property rights system that stimulates innovation and reduces risks of innovation, including private property rights, patent systems and intellectual property protection systems, is the key to the sustainability of technological innovation.

Based on the existing research findings, it is suggested that both technological and institutional instruments should be included in measuring regional innovation capacity. In addition, the role of the coordination level between technological and institutional innovation in promoting green growth was tested as well. It is expected that both technology and institution create incentives.  Thus, the following hypothesis in this paper is highlighted:

H3a: The performance of green growth is significant and positively affected by the combination of technological instruments and institutional instruments.

H3b: The level of coordination between regional technological innovation capacity and institutional innovation capacity has a positive effect on green growth performance.

It is well-known that technological innovation directly affects economic growth. The effects are from the developing green and clean technology to realizing the recycling of resources, optimizing the energy consumption structure, and improving the utilization rate of resources. While institution and systemic innovation aims to connect different innovators, such as universities, scientific research institutions, and enterprises, and encourage knowledge and technology learning and resource sharing by building knowledge exchange and technology transfer platforms, it includes infrastructure construction and encourages various forms of technical cooperation among innovators [54][55]. The combination of these two types of innovation capacity constitutes the regional innovation system, which needs to be consistent to maintain the long-term sustainability of green growth.

3. Response to comment: We are very sorry for our incorrect writing, and have made correction according to the Reviewer’s comments as below:

Actions: the statements of lines 252 is correct as “ As shown, the trend is almost the same in these three types of RIC, while there is sharp rise in soft curve after 2014 and a dramatic fall in 2015”.

4. Response to comment: As Reviewer suggested that Author’s calculations should based on some datat. We have made correction according to the Reviewer’s comments as below:

Actions: the statement is replaced by “Source: Author’s calculations on China’s Year Book”

5. Response to comment: We are very sorry for our incorrect writing, and have made correction according to the Reviewer’s comments as below:

Actions: the statements of lines 252 is correct as “ This paper examines the effects of regional innovation capacity and its internal characteristics on green growth in a cross-section of China’s provinces over the period 2008–2016.”

Special thanks to you for your good comments. 

Round 2

Reviewer 1 Report

Accept in present form